# On-surface synthesis of triangulene trimers via dehydration reaction

Suqin Cheng[1,4], Zhijie Xue[1,4], Can Li [2,4], Yufeng Liu[2], Longjun Xiang[1], Youqi Ke[1], Kaking Yan [1], Shiyong Wang [2,3✉] & Ping Yu [1✉]

Triangulene and its homologues are of considerable interest for molecular spintronics due to their high-spin ground states as well as the potential for constructing high spin frameworks. Realizing triangulene-based high-spin system on surface is challenging but of particular importance for understanding $\pi$-electron magnetism. Here, we report two approaches to generate triangulene trimers on Au(111) by using surface-assisted dehydration and alkyne trimerization, respectively. We find that the developed dehydration reaction shows much higher chemoselectivity thus resulting in significant promotion of product yield compared to that using alkyne trimerization approach, through cutting the side reaction path. Combined with spin-polarized density functional theory calculations, scanning tunneling spectroscopy measurements identify the septuple ($S = 3$) high-spin ground state and quantify the collective ferromagnetic interaction among three triangulene units. Our results demonstrate the approaches to fabricate high-quality triangulene-based high spin systems and understand their magnetic interactions, which are essential for realizing carbon-based spintronic devices.

[1] School of Physical Science and Technology, ShanghaiTech University, 201210 Shanghai, China. [2] Key Laboratory of Artificial Structures and Quantum Control (Ministry of Education), Shenyang National Laboratory for Materials Science, School of Physics and Astronomy, Shanghai Jiao Tong University, 200240 Shanghai, China. [3] Tsung-Dao Lee Institute, Shanghai Jiao Tong University, 200240 Shanghai, China. [4]These authors contributed equally: Suqin Cheng, Zhijie Xue, Can Li. ✉email: shiyong.wang@sjtu.edu.cn; yuping@shanghaitech.edu.cn

Triangulene due to its unique triangular zigzag non-Kekulé structure is one of the most fundamental open-shell $\pi$ conjugated nanographenes with high-spin ground state, particularly useful for carbon-based spintronic devices[1,2]. The ground state spin quantum number of triangulene can be quantified using Ovchinnikov's rule and Lieb's theory for bipartite honeycomb lattice[2,3], $S = (N_A - N_B)/2 = 1$, where $N_A$ and $N_B$ represent the number of carbon atoms of the two interpenetrating sublattices (Fig. 1a). Although derivatives of triangulenes have been obtained in solution[4–6], synthesis of unsubstituted triangulenes has been a great challenge for the conventional wet-chemistry method because of the high reactivity arising from their unpaired electrons. In this respect, on-surface synthesis under ultrahigh vacuum has appeared as a powerful approach for the synthesis and characterization of such highly reactive (open shell) nanographenes. Upon tip-assisted synthesis or appropriate molecular precursor design, unsubstituted triangulene[7] and its larger homologs, [4]-, [5]- and [7] triangulene molecules[8–10] have been successfully fabricated on different surfaces and their chemical as well as the electronic structures have been characterized using scanning probe techniques with sub-molecular precision.

In addition to obtaining large homologous series of [n]triangulenes, on-surface synthesis also holds great promise in the fabrication of extended triangulene-based nanostructures via appropriate design of molecular precursors[11–13]. It can provide manifold nanostructures with tailored spin quantum number and magnetic ordering which are crucial for both fundamental and technological prospects[14–17]. In recent years, trianglulene based nanostructures such as the triangulene dimers[18], triangulene quantum ring[19] as well as triangulene-based molecular chain[20] and nanorings[21] have been obtained on metal surfaces via on-surface synthesis. However, the magnetic orderings in reported triangulene-based nanostructures are limited to the antiferromagnetic coupling, since the neighboring triangulene units are connected through two carbon atoms from different sublattices via direct carbon-carbon addition coupling. In this case, as illustrated in Fig. 1b, the number of carbon atoms in A sublattice is exactly the same as that in B sublattice, and thus the resulted

triangulene dimers are with a net spin of $S = 0$ according to Lieb's theorem. To fabricate the triangulene-based high-spin system, the triangulene units are required to be connected at the same sublattice sites of a benzene or a triazine spacer (Fig. 1c) for generating sublattice imbalance and thus the high-spin ground state, which remains as a challenging task for both in-solution and on-surface synthesis.

In this regard, formation of additional benzene or triazine ring on surfaces is of particular interest because it allows for realizing triangulene-based high-spin frameworks. As the reported investigations, trimerization of rationally designed precursor with alkynes terminals on the surface are highly promising for generating an additional benzene linker[22,23]. The shortcoming is that different reaction pathways of alkyne functional group such as homocoupling and cross-coupling can proceed concurrently resulting in the products diversity[24–28]. On the other hand, cyclotrimerization of nitrils has been widely applied in solution for incorporating additional triazine ring[29]. However, the formation of additional triazine ring on surfaces has remained challenging[30,31]. In this work, we report on-surface synthesis of covalently bonded high-spin triangulene trimers via a triazine ring through surface-assisted dehydration reaction. Particularly, we find this approach has fewer side reactions and can increase the product yield markedly compared to the trimerization of precursors with alkynes terminals. Supported by spin-polarized density functional theory calculations, the septuple ($S = 3$) high-spin ground state and the magnetic coupling strength are evidenced by scanning tunneling spectroscopy (STS) results. Our results provide a promising approach to obtain high-spin triangulene-based system with high yield and quality, which is a crucial step for realizing triangulene-based two-dimensional high-spin networks.

## Results

**On-surface synthesis of triangulene trimers linked by a benzene ring.** In order to fabricate the covalently bonded triangulene trimers via a benzene ring (Tb) as shown in Fig. 1c, alkyne trimerization of precursor 9-(4-ethynyl-2,6-dimethylphenyl)anthracene

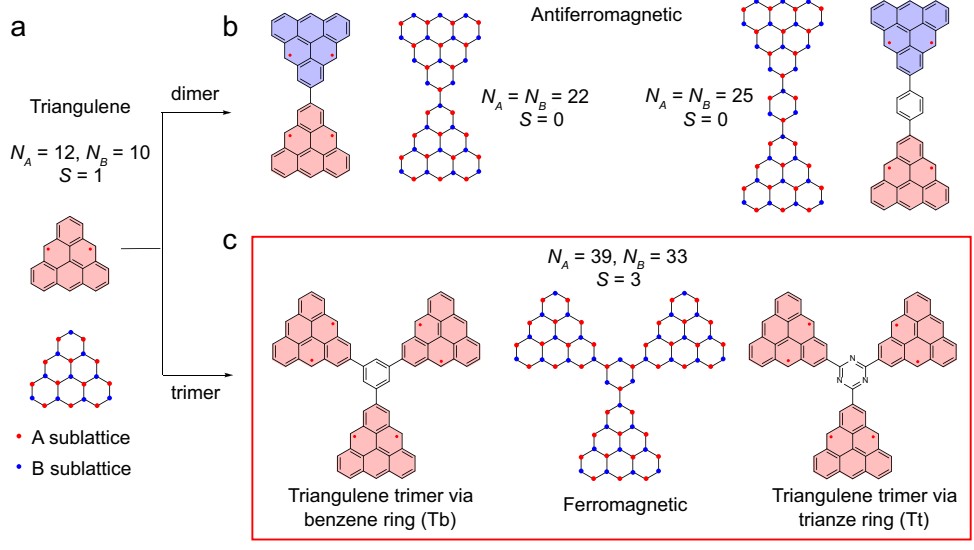

**Fig. 1 Magnetic coupling in triangulene dimers and trimers. a** Chemical structure of triangulene with unpaired electrons. Two interpenetrating triangular sublattices highlighted as blue and red filled circles giving $S = 1$ magnetic ground state. The symbols A and B are represented as carbon atoms of two interpenetrating sublattices in the nanographene structure. **b** Schematic drawings for two kinds of triangulene dimers with antiferromagetic ordering. **c** Schematic drawings for triangulene trimers connected with benzene or triazine ring. The net sublattice imbalance of six in these structures give rise to $S = 3$ high-spin ground state. The filling colors in triangulenes denote their ferromagnetic or antiferromagnetic couplings.

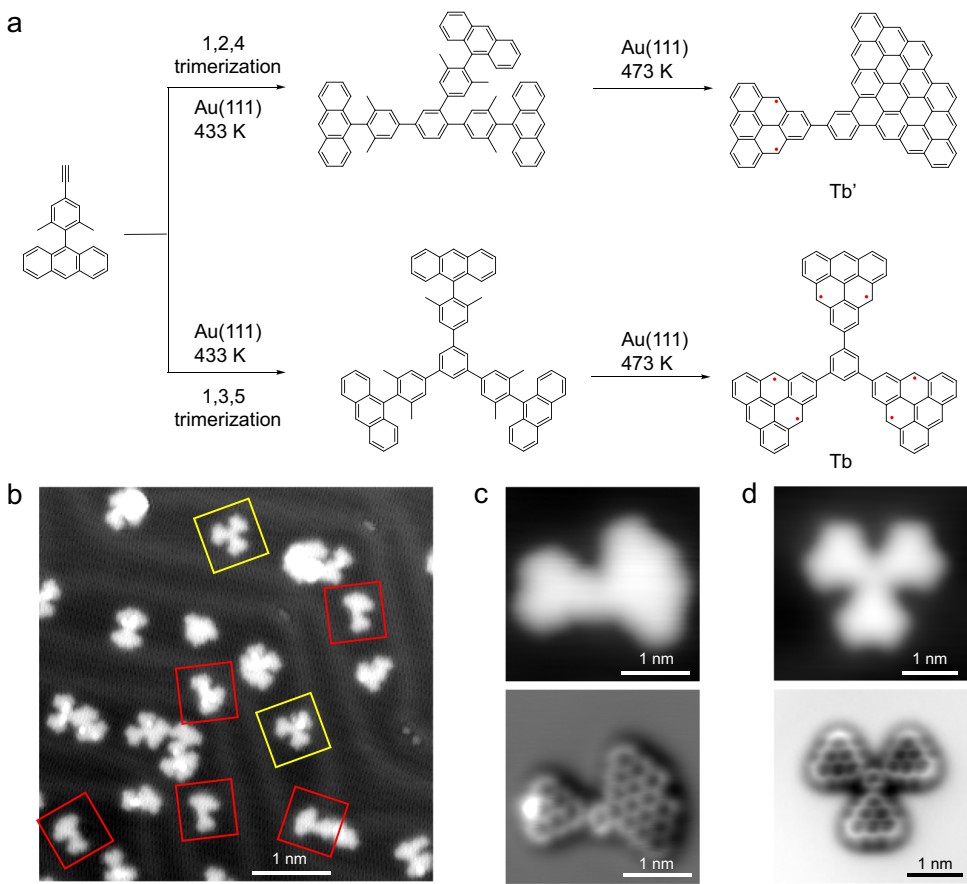

**Fig. 2 Chemical structure characterization of products via trimerization of precursors with alkynes terminals. a** Possible on-surface reactions for the main products. **b** Overview STM image after annealing precursor on Au(111) at 433 K and 473 K sequentially. **c, d** Zoom-in STM and AFM images of two main products on Au(111) with CO tip. Setpoint: $V = 300$ mV, $I = 30$ pA; $\Delta z = 0.02$ nm.

(Supplementary Fig. 2) was tried on a Au(111) surface (Fig. 2a). Although trimerization reaction can happen, Tb was only be obtained with very low yield of 2.3% as illustrated in the overview STM image of Fig. 2b (nearly 2000 molecules are counted of different overviews for statistics). Instead, the main products are Tb' (yield of 11%) as marked in Fig. 2b with red frames. In order to characterize the chemical structures of the products precisely, non-contact atomic force microscopy (nc-AFM) measurements were performed in the constant-height mode with a CO-functionalized tip (Fig. 2c, d)[32,33]. Figure 2c, d show the high-resolution STM and AFM images of Tb' and Tb, exhibiting triangulene linked with a rhombus-shape nanographene and triangulene trimers linked via a benzene ring respectively. The bright spot in the AFM image of Fig. 2c is due to an extra hydrogen passivation. Due to the radical character, the carbon sites with unpaired electrons of triangulene show reactivity and can get passivated randomly by hydrogen atoms produced during dehydrogenation reaction on the surface. The observed triangulene trimers with hydrogen passivation at different carbon sites can be found in Supplementary Fig. 23. As the reaction path shown in Fig. 2a, the precursors are cyclotrimerized after annealing at 433 K, forming benzene rings with two kinds of isomers, symmetric 1,3,5- and asymmetric 1,2,4-trisubstituted products. From the statistical analysis of the STM images, it suggests that the 1,2,4-isomeric structure is more favorable to be formed in the trimerization process, which is consistent with the theoretical probability ratio of 3:1 for forming unsymmetric 1,2,4-isomeric structure to symmetric 1,3,5-structure. To avoid forming 1,2,4-isomer reaction path and promote the yield of trianglulene trimers, we tried to take the advantage of cyclotrimerization of nitrils reported in solution[29], the linking unit of

benzene ring can be replaced with 1,3,5-triazine ring as illustrated below.

**On-surface synthesis of triangulene trimer via a triazine ring upon dehydration reaction.** We first tried to synthesize precursor **2** (Fig. 3a) via solution-mediated cyclotrimerization. However, the solution-mediated cyclotrimerization only led to the formation of a dimer product **3**[34], which is confirmed by NMR, mass spectra and XRD (Supplementary Figs. 6–8). Then, for obtaining **2**, we attempted to co-deposit dimer **3** and precursor **4** with nitril terminal on Au(111) surface. We have tried different annealing temperatures and found that trimer **2** can be successfully synthesized through surface-assisted dehydration reaction on Au (111) surface with maximized yield upon co-deposition of **3** and **4** then annealing to the temperature of 453 K for 5 min. The other key parameter should be taken into account is the amount ratio of dimer **3** to monomer **4** used in co-deposition. In principle, for dehydration reaction the ratio of **3** to **4** is required as 1:1. However, considering the desorption temperature of monomer is lower than that of dimer, the excess amount of monomer **4** in the evaporation will enhance the possibility for dehydration reaction. Indeed, we find that upon using excess **4** in the co-evaporation, triangulene trimers can be obtained with a high reaction yield of 60% for both low and high coverages, as shown in the overview STM images of Fig. 3b, c (nearly 2000 molecules are counted of different overviews at low and high coverages for the statistics). By further annealing the substrate at 473 K, **2** underwent cyclodehydrogenation reaction,

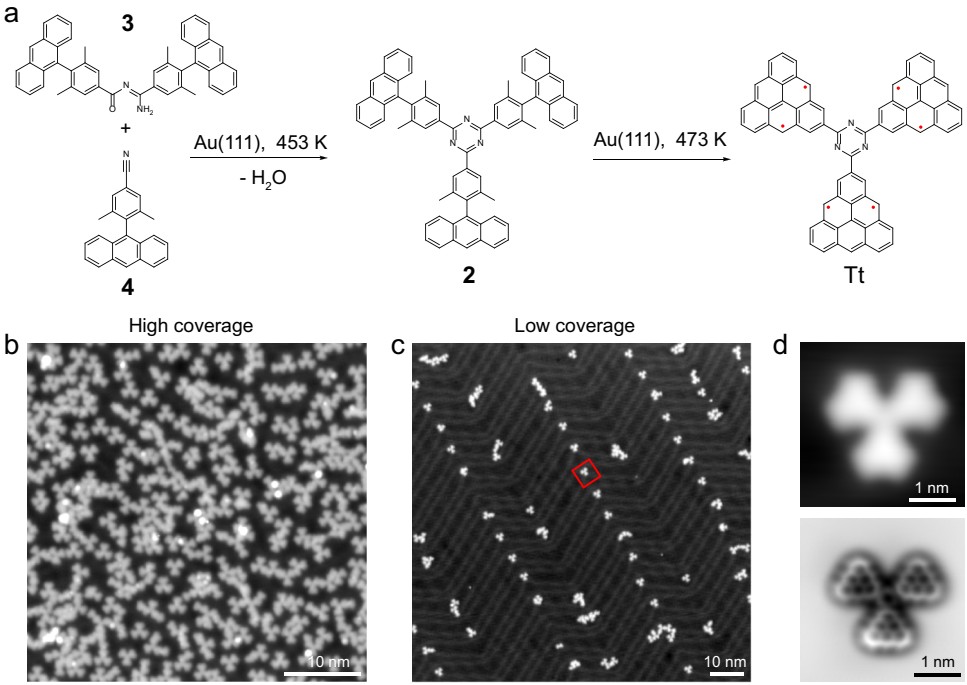

**Fig. 3 Synthesis and chemical structure characterization of triangulene trimers Tt. a** On-surface synthetic routs toward structure Tt. **b, c** Overview STM images after annealing precursors **3** and **4** on Au(111) at 453 K at high and low coverages. **d** Zoom-in STM and AFM images of Tt on Au(111) with CO-functionalized tungsten tip. Setpoint: $V = 300$ mV, $I = 30$ pA; $\Delta z = 0.02$ nm.

and mostly transformed to the well-isolated final target Tt. The corresponding STM and nc-AFM images shown in Fig. 3d unambiguously resolve the molecular backbone consisting of three triangulenes linked with a ring spacer, confirming the successful fabrication of the expected Tt. Its flat adsorption geometry also indicates there is no chemical bonding to the substrate.

**Electronic structures of triangulene trimers Tt and Tb.** As per Lieb's theorem, triangulene trimers connected with a benzene or trianze ring should both have high-spin state of $S = 3$ due to sublattice imbalance, where the bipartite lattice hosts 39 atoms in one sublattice and 33 atoms in the other. Mean-field Hubbard model (MFHM) and spin-polarized density functional theory (SP-DFT) have been used to calculate the electronic properties of Tb and Tt. Both methods indicate that $S = 3$ is the ground state for both Tb and Tt, predicting a ferromagnetic ordering among three triangulene units, which is agree with Lieb's theory. The MFHM results can be found in Supplementary Fig. 16. The DFT calculations reveal that the ground state $S = 3$ is with a lower energy of 13.9 meV and 8.9 meV than that of the first excited $S = 1$ state for Tb and Tt, respectively. Figure 4a, b depict the calculated energy levels of Tb and Tt in the ground state $S = 3$. Their frontier orbitals both contain six singly occupied and singly unoccupied molecular orbitals (SOMOs and SUMOs). The quasi-degenerated SOMOs are populated by six ferromagnetically coupled electrons, giving the ground state of $S = 3$. The orbital shapes of SOMOs are shown in Fig. 4c, d. To experimentally compare the electronic structure of Tb and Tt, we measured differential conductance (d$I$/d$V$) spectra at different positions over Tb/Tt using a CO-functionalized tungsten tip. As Fig. 4e, f shown, the d$I$/d$V$ spectra measured at the edge corner of Tb/Tt (measured position marked by red dot in the inset image) show two pronounced peaks at $-0.58$ V/$-0.5$ V and 0.93 V/1.35 V for Tb and Tt, respectively. These two peaks could be attributed to the presence of the SOMOs and SUMOs. In addition, two shoulder features around $-1.4$ V/$-1.8$ V and 1.8 V/2 V observed

in d$I$/d$V$ spectra measured at the connection corner on Tb/Tt could be attributed to several occupied molecular states with close energy (H1) and several unoccupied molecular states (L1), while the substantial rise above $-2.0$ V may originate from other higher occupied molecular states H2 (The assigned states are marked by dashed rectangle in Fig. 4a, b).

To further resolve the spatial distribution of these molecular orbitals, we have performed d$I$/d$V$ mappings at the observed resonant energies. As the d$I$/d$V$ maps measured at $-0.5$ V and 1.35 V on Tt shown (Fig. 5a), they are similar and exhibit characteristic nodal patterns localized at the zigzag edges of triangulenes, which agree well with the simulated local density of states (LDOS) maps obtained by superimposing the six orbitals of SOMOs and SUMOs. The d$I$/d$V$ map obtained at 2 V shows three bright spots in the center, which fit with the simulated LDOS map obtained by superimposing the molecular states of L1 as labeled in Fig. 4a. The d$I$/d$V$ maps acquired at $-1.8$ V and $-2.3$ V show faint nodal features and triangle frame along the zigzag edges, which can be reproduced in the simulated LDOS maps using occupied molecular states H1 and H2 (H1 and H2 are marked in Fig. 4a). Similarly, the d$I$/d$V$ maps measured at different resonance energies of Tb are also compared with theoretical simulations, which show similar wave patterns to that of Tt (Supplementary Fig. 17). Therefore, we can conclude that both benzene and trianze linkers give the ground state of $S = 3$, that is, three triangulene units are ferromagnetic coupled. The electronic structures of the excited spin state $S = 1$ for Tb and Tt can be found in Supplementary Figs. 18, 19.

**Spin flip spectroscopy of triangulene trimers.** To manifest the magnetic state, d$I$/d$V$ spectrum was measured near the Fermi level as shown in Fig. 5b[35,36], which shows symmetric peaks at 5 meV respectively. The absence of Kondo resonance has also been found for [4]- and [5]-triangulenes with high-spin quantum numbers[8,9], since the observation of Kondo resonance of such high-spin system should be challenging due to its partial screening character. These low-energy resonances are too weak to be detected at a larger setting

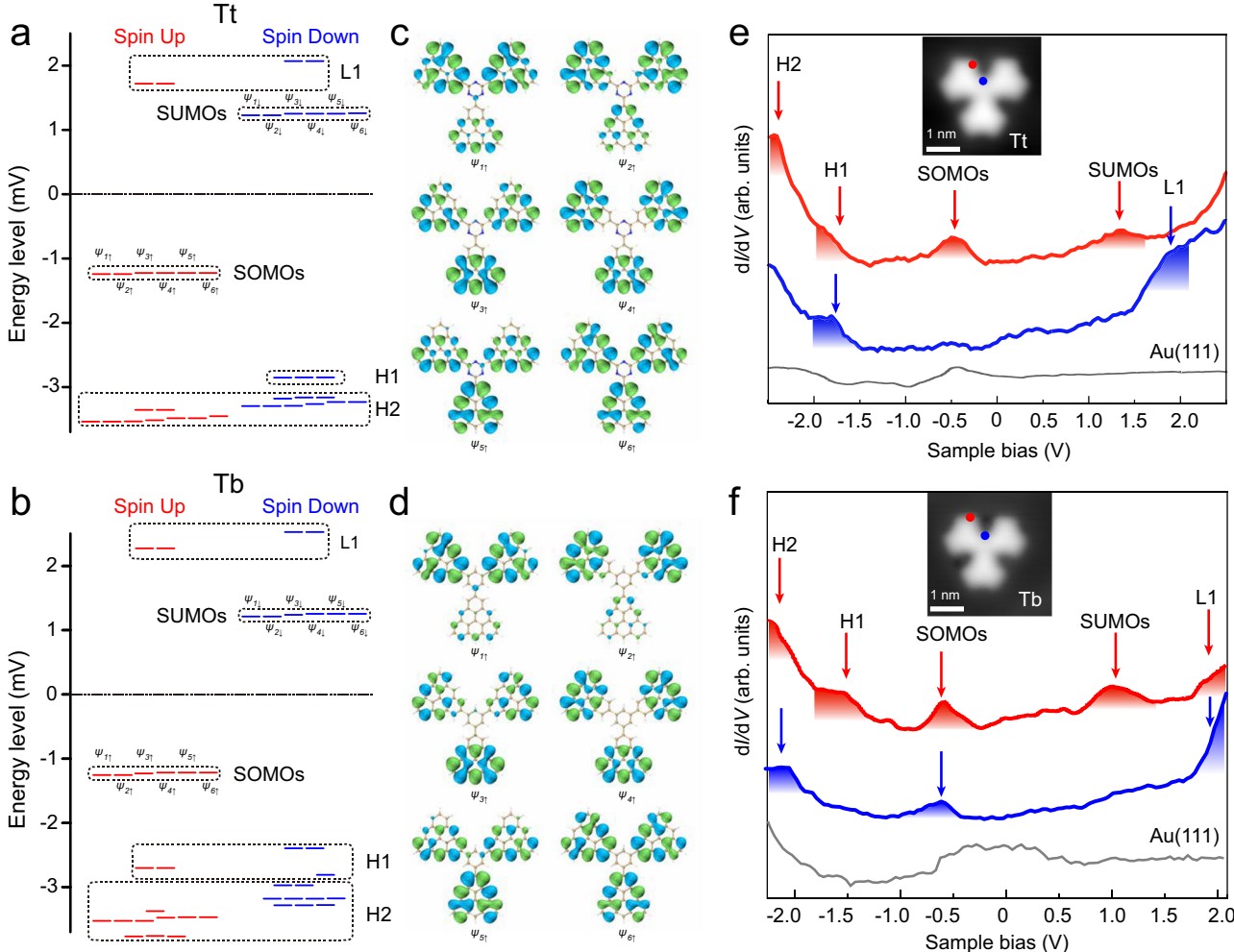

**Fig. 4 Comparison of electronic structures between Tt and Tb. a**, **b** DFT-calculated spin-polarized energy diagram of Tt and Tb. **c**, **d** DFT-calculated wave functions of six SOMOs. The blue and green isosurface colors indicate opposite phases of the wave function. **e**, **f** d$I$/d$V$ spectra on Tt and Tb measured at the positions as marked with filled circles in the inset images.

position as Fig. 4e, thus requiring a close setting point (tip-sample separation) for low-energy d$I$/d$V$ measurement. The d$I$/d$V$ maps recorded at these resonant energies (Fig. 5c, d) show a similar spatial distribution as the d$I$/d$V$ maps of SOMOs/SUMOs (Fig. 5a). Comparing the calculated energy levels of $S = 3$ and $S = 1$ (Supplementary Fig. 20), some levels slightly shift with energy differences around few meV. Such difference cannot be distinguished in experiments by comparing the experimental d$I$/d$V$ maps with the theoretical simulated LDOS maps of $S = 3$ and $S = 1$ states (Supplementary Fig. 21). To ambiguously determine the ground state in the experiment, spin-polarized measurements are required for obtaining the spin density contrast, which is beyond the scope of this work. However, based on both the spin-resolved DFT and mean-field Hubbard calculation results, $S = 3$ is the ground state with an energy around 10 meV lower than that of $S = 1$. Therefore, according to the calculation results, we assign the ground state to $S = 3$ and attribute this spin-flip signature[37] to the transition from the ground state $S = 3$ to the first excited state of $S = 1$. The measured magnetic coupling strength is about 10 meV, which also agrees well with theoretical predictions (Fig. 5e). The results for the spin excitation of Tb can be also found in Supplementary Fig. 17.

## Discussion

In summary, triangulene trimers linked via a benzene/1,3,5-triazine ring have both been successfully fabricated through different on-surface reactions. Moreover, we report the dehydration reaction, which can incorporate additional trianze ring on surface and promote the high yield of triangulene trimers by avoiding the side reaction path of forming 1,2,4-isomeric structures. Their chemical and electronic structures are precisely characterized by nc-AFM imaging and d$I$/d$V$ measurements. Combined with theoretical calculations, its septuple ($S = 3$) ground state is identified through investigating the molecular resonances and low-energy spin excitations. Our results provide insights to fabricate the triangulene-based high-spin systems, which are inspired for both fundamental and technological applications.

## Methods

**Sample preparation and details of STM measurements**. The sample preparation is carried out under UHV conditons ($7 \times 10^{-10}$ mbar). Au(111) single-crystal was cleaned by cycles of $Ar^+$ ion sputtering and subsequently annealing to 773 K. The precursors were thermally evaporated onto the clean Au(111) surface through co-evaporation of monomers and dimers at room temperature. All the STM/AFM experiments were performed at 5 K with commercial Createc LT-STM/qPlus AFM. The qPlus sensor with a resonance frequency of 30.4 KHz, the oscillation amplitude of 50 pm. After dosing CO, we picked up CO molecule from the Au(111) surface by increasing the current to 0.1 nA to prepare the CO-functionalized tungsten tip. For the constant-height AFM images, the tip-sample distance is decreased a few hundred pm from the STM set point $V = 300$ mV, $I = 30$ pA. The d$I$/d$V$ spectra were measured by using a lock-in amplifier (515 Hz, 20 mV, or 0.1 mV modulation). The STM images, d$I$/d$V$ spectra and d$I$/d$V$ maps are also measured with a CO-functionalized tungsten tip.

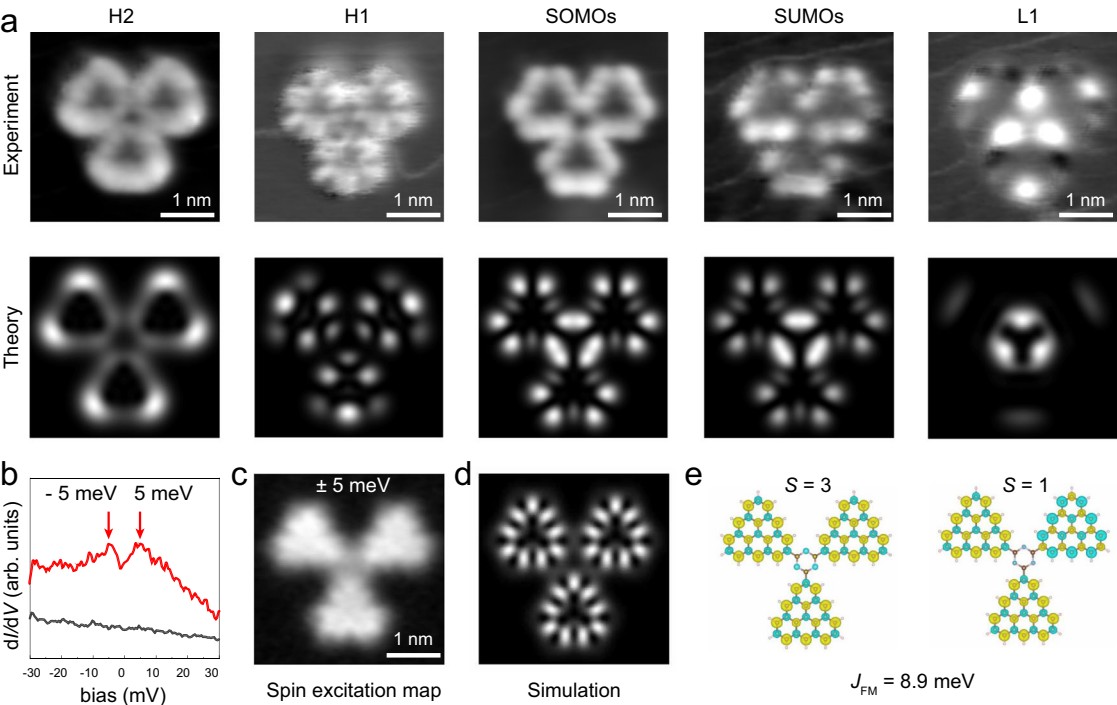

**Fig. 5 Experimental and simulated dI/dV maps of Tt. a** Constant-current dI/dV maps (top) and theoretical simulated maps (bottom) at energies of −2.3 V (H2), −1.8 V (H1), −0.5 V (SOMOs), 1.35 V (SUMOs) and 2 V (L1). ($V_{rms} = 25$ mV, $I = 200$ pA). **b** Low-energy dI/dV spectra on Tt. **c**, **d** Constant-height dI/dV map (**c**) and theoretical simulated map (**d**). **e** DFT-calculated spin density distribution of Tt in spin state of $S = 3$ and $S = 1$. The colors green and yellow represent the spin density distributions. Set point: (**b**) $V = 30$ mV, $I = 200$ pA; (**c**) $V_{rms} = 2$ mV.

**Density functional theory simulations**. Spin-polarized DFT calculations were performed with Gaussian 16. The ground state structures of gas-phase Tb and Tt were optimized by the PBE0-D3 (BJ)[38–40] functional combined with the def2-SVP basis set[41], which was extended to a def2-TZVP basis set[41] for the single point energy calculation. The Tb structure, in particular, was forced to be planar in studies to match the physical adsorption on substrate, ignoring the deformation caused by steric hindrance of hydrogen atoms between the benzene ring and the triangulenes. Molecular orbitals and electron spin densities were analyzed by Multiwfn[42]. Images of the structures and isosurfaces were plotted using VESTA[43] and VMD[44].

**Mean-field Hubbard calculations**. The tight binding (TB) calculation of the STM images was carried out in the C $2p_z$-orbital description by numerically solving the Mean-Field-Hubbard Hamiltonian with nearest-neighbor hopping:

$$\hat{H}_{MFH} = \sum_{\langle ij \rangle, \sigma} -t_{ij} c_{i,\sigma}^{+} c_{j,\sigma}^{-} + U \sum_{i,\sigma} n_{i,\sigma} n_{i,\overline{\sigma}} - U \sum_i n_{i,\uparrow} n_{i,\downarrow} \quad (1)$$

with $t_{ij}$ is the nearest-neighbor hopping term depending on the bond length between C atoms (For simplicity, we choose $t_{ij} = 2.7$ eV), and $c_{i,\sigma}^{+}$ and $c_{j,\sigma}^{-}$ denoting the spin selective ($\sigma = \uparrow, \downarrow$) creation and annihilation operators on the atomic site $i$ and $j$, $U$ the on-site Hubbard parameter (with $U = 3.5$ eV used here), $n_{i,\sigma}$ the number operator and $n_{i,\sigma}$ the mean occupation number at site $i$. Numerically solving the model Hamiltonian yields the energy Eigenvalues $E_i$ and the corresponding Eigenstates $\alpha_{i,j}$ (amplitude of state $i$ on site $j$) from which the wave functions are computed assuming Slater type atomic orbitals:

$$\psi_i(\vec{r}) = \sum_j \alpha_{i,j} \cdot (z - z_j) \exp(-\zeta |\vec{r} - \vec{r_j}|) \quad (2)$$

with $\zeta = 1.625$ a.u. for the carbon $2p_z$ orbital. The charge density map $\rho(x,y)$ for a given energy range $[\varepsilon_{min}, \varepsilon_{max}]$ and height $z_0$ is then obtained by summing up the squared wave functions in this chosen energy range.

$$\rho(x,y) = \sum_{i,\varepsilon_i \in [\varepsilon_{min}, \varepsilon_{max}]} \psi_i^2(x, y, z_0) \quad (3)$$

Constant charge density maps are taken as a first approximation to compare with experimental STM images.

## Data availability
All data generated in this study are available within the article and supplementary information, or from the corresponding authors upon request. Source data are provided with this paper. Crystallographic data for this paper can be obtained free of charge from the Cambridge Crystallographic Data Center via www.ccdc.cam.ac.uk/data_request/cif under the deposition number 2107270. Source data are provided with this paper.

## Code availability
The tight-binding calculations were performed using a custom-made code on the MATLAB platform. Details of this tight-binding code can be obtained from the corresponding author on request.

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

## Acknowledgements

The author S.W. acknowledges the financial support from the National Key R&D Program of China (No. 2020YFA0309000), the National Natural Science Foundation of China (No. 11874258, No. 12074247), the Shanghai Municipal Science and Technology Qi Ming Xing Project (No. 20QA1405100), Fok Ying Tung Foundation for young researchers and SJTU (No.21X010200846). P.Y. gratefully acknowledges the financial support from the Science and Technology Commission of Shanghai Municipality (20ZR1436900) and ShanghaiTech start-up funding.

## Author contributions

S.Q.C. and P.Y. conceived the experiments. P.Y. and S.W. supervised the project. S.Q.C. designed and synthesized the molecular precursors. Z.J.X. measured the experimental data. C.L. and L.Y.F. performed the DFT and MFHM calculations. L.J.X, Y.Q.K., and K.K.Y. provide support during the theoretical calculations and molecular synthesizing. All the authors discuss the results and commented on the manuscript. S.Q.C, Z.J.X, and C.L. contributed equally to this work.

## Competing interests

The authors declare no competing interests.
