## [Peer Review File · Nature Communications]

On-Surface Synthesis of Triangulene Trimers via Dehydration ReactionREVIEWER COMMENTS

Reviewer #1 (Remarks to the Author):

Please find an assessment of the manuscript no. NCOMMS-21-37213 presented by S. Cheng, et al entitled as "On-Surface Synthesis of Triangulene Trimers via '2+1' Dehydration Reaction". The article presents an interesting study reporting on-surface synthesis of triangulene-based high-spin molecular system with ferromagnetic coupling. Recently, magnetic carbon-based pi-electron systems have received a lot of attention, not only because of some applications in spintronics but it has also relevance of basic understanding of electronic structure of open-shell polyaromatic hydrocarbons. This paper represents an interesting contribution to this field.

I found the synthetic part very interesting with nice ideas how to overcome the problem of low reaction yield of original precursor 4 via trimerization process. I would appreciate more detailed description of statistical analysis of the yield of reaction, showing large STM images and providing information about data static (how many molecules were counted, coverage and temperature dependence). I would also appreciate some more elaborated comments on possible reaction mechanism, role of adatom/substrate with possible. But these are minor comments.

On the other hand, I found theoretical analysis very weak and careless. The authors rationalise experimental dI/dV maps and spectra using Hubbard mean field method, which depends strongly on employed parameters U/t. It is not clear, why parameter U was chosen 3.5 eV (it seems a small value for carbon/nitrogen) and considered the same for both N and C atoms? One should not rely on a relatively good match with experimental dI/dV maps. Instead, one should try to understand better the magnetic ground and excited states providing a comparative analysis of different U/t parameters. Does it mean that the triazine linker does not have any influence on the magnetic structure and spin excitation spectra? Indeed, what is missing completely is a detailed discussion of possible difference between electronic structure of Tb/Tt products. I suggest to carry out at least some fully relaxed DFT gas-phase calculations with hybrid XC to analysis the atomic and electronic structure of both products including both the ground and first excited states. Such analysis could provide more information about the differences as well as more insight into the effect of triazine linker on the magnetic states and spin excitations. I convinced that these information will be appreciated by colleagues working in the same field and it will extend our understanding of pi-electron magnetism in carbon-based systems.

In conclusions, I think this is an interesting paper which may deserve publication in Nature Comm, after careful revision and incorporation of the comments above.

Reviewer #2 (Remarks to the Author):

The manuscript describes the synthesis of high-spin triangulene trimers on Au (111) surface formed by alkyne trimerization and a novel 2+1 surface-assisted dehydration, and characterization of these trimers by AFM and STM techniques. The obtained results are supported by DFT calculations. The novelty of the presented manuscript lies in the connection of triangulene units via benzene/triazine linker, which creates sublattice imbalance resulting in a high-spin ($S = 3$) system. Presented STM and low-temperature

dI/dV spectra support conclusions made by authors about the ground state of the triangulene trimer. However, I am not an expert in the field of on-surface spectroscopy and a reviewer from the field should judge whether the presented STM results suffice to confirm the ground state and antiferromagnetic behavior of the presented molecule.

The discussion of the results obtained for triangulene trimer linked via benzene is omitted from the discussion. In my opinion, it would be helpful for the readers of this paper to add a paragraph describing these results, including a direct comparison with the results for trimer with a triazine linker and making a conclusion about the effect of heteroatoms.

Overall, I think that this work possesses novelty and quality of research required for Nature Communications. However, this manuscript contains significant technical issues. After a revision addressing all the points listed below, this manuscript can be re-considered for publication in Nature Communications.

Comments:

- 1) Compounds 4, 5, 6 are prepared for the first time (no result in Reaxys&SciFinder). Therefore they must be characterized by another analytical method than just NMR, namely, HR-MS or elemental analysis.
- 2) In the synthetic procedure for preparation of compound 4, K_2CO_3 is above the reaction arrow but K_3PO_4 in the text.
- 3) NMR is not correctly described several times. For example: 1) For compound 5, the singlet for 2H from phenyl ring is reported as a doublet at 7.41 (based on the amount of H), but this signal cannot be described as a doublet because it is a mixture of 2 signals. Moreover, the coupling $J = 12.7$ Hz does not fit other coupling constants. 2) For compound 3, methyl groups are described as a doublet with $J = 7.5$ Hz, but it should be described as two independent singlets. All mistakes of this kind should be corrected.
- 4) What are the impurities in the ^{13}C NMR of 5 (Fig. S7)? Authors should assign these impurities.
- 5) Page 3, Figure 1: Unpaired electrons are part of the π system. I suggest to draw them from the inside of the rings.
- 6) Page 4, line 73: Authors use abbreviation Tbs, which does not belong to any compound (probably a typo).
- 7) Page 5, Figure 2c: Authors assign bright spot to hydrogen passivation. Why does it happen only with this hydrogen and not with others? A more detailed description in the main text would be helpful.
- 8) Page 6, line 90: Authors use Fig 3 instead of Fig 3a.
- 9) Page 8, Figure 4d: How did the authors assign the energy of HOMO? There is no pronounced peak at this energy. It looks like the peak for SOMO is composed of 2 peaks. What is the second peak?
- 10) Figure 2 + Figure 4: It would be clearer to add a size of the scale bar right to the picture instead of having it in the description of the figure.

Reviewer #3 (Remarks to the Author):

Suqin Cheng et al. report on the on-surface synthesis of triangulene trimers via a 2+1 dehydration reaction. They compare reaction schemes yielding triangulene trimers that are connected via a single benzene or triazine ring. These triangulene trimers (Tb and Tt, for benzene and triazine linker, respectively) are exciting as they are supposed to realize nanographenes with high-spin ground states (the connection geometry results in a sublattice imbalance, which implies a certain spin-state via Lieb's theorem), namely six unpaired

electrons and $S=3$ in the present case. The synthesis approach is novel, the experiments are carefully carried out and the results are exciting for the broad community working on carbon-based (nanographene) magnetism. I recommend publication in Nature Communications once my comments below have been answered.

1. The authors should compare the experimental dI/dV maps with the theoretically predicted maps also for the other possible spin states (namely $S=1$). This would give more confidence to the assignment of the $S=3$ spin state.

2. The data in Figure 4: experimental orbital maps were carried out under constant-current conditions, which means that part of the contrast results from the varying tip height. This has minor effect for SOMO and SUMO, but is visible for the higher lying orbitals. Did the authors carry out constant-height experiments and are the theory predictions simulated for constant-current or -height? In addition, how does the experimental dI/dV map of HOMO-1 look like?

3. The spin-flip transitions (Fig. 4f): what transition does it correspond to and what is the predicted energy of this? How would this change if the ground state was $S=1$?

4. What are the differences in the theory predictions for Tt and Tb molecules? Presently, the manuscript silently implies that all the spin transitions are identical. I understand that the ground state should be $S=3$ in both cases, but what about the predicted spin-flip energies? Experimentally they seem very similar.

5. Do all the SOMO and SUMO orbitals have identical energy or are there some small splittings (according to the calculations)?

Minor:

- Please add experimental parameters for the AFM experiments, e.g. cantilever oscillation amplitude. Secondly, I wanted to check if the STM experiments are carried out with a CO terminated or a metallic tip?

Response to Reviewer 1:

Comments: Please find an assessment of the manuscript no. NCOMMS-21-37213 presented by S. Cheng, et al entitled as "On-Surface Synthesis of Triangulene Trimers via '2+1' Dehydration Reaction". The article presents an interesting study reporting on-surface synthesis of triangulene-based high-spin molecular system with ferromagnetic coupling. Recently, magnetic carbon-based pi-electron systems have received a lot of attention, not only because of some applications in spintronics but it has also relevance of basic understanding of electronic structure of open-shell polyaromatic hydrocarbons. This paper represents an interesting contribution to this field.

We thank the reviewer for the positive comments. We have performed additional experimental and theoretical results to address the following comments/concerns and modified our manuscript accordingly.

1. I found the synthetic part very interesting with nice ideas how to overcome the problem of low reaction yield of original precursor 4 via trimerization process. I would appreciate more detailed description of statistical analysis of the yield of reaction, showing large STM images and providing information about data static (how many molecules were counted, coverage and temperature dependence). I would also appreciate some more elaborated comments on possible reaction mechanism, role of adatom/substrate with possible. But these are minor comments.

We thank the reviewer's nice suggestions, which will certainly improve our manuscript. We performed additional experiments with different annealing temperature ranging from 393 K to 473 K. We find that 453 K is the best annealing temperature for maximizing the yield of triangulene trimers. Using the temperature below 453 K, the '2+1' dehydration reaction can't take place fully. If the annealing temperature is too high, the monomer precursor will desorb quickly, which also gives low yield of trimers.

Another key parameter to get high yield of triangulene trimers is to co-deposit monomer and dimer simultaneously and choose proper ratio of dimer to monomer. Although the ideal ratio of dimer to monomer is 1:1 for '2+1' dehydration in principle, excess monomer will enhance the possibility of '2+1' dehydration reaction since the desorption temperature of monomer is lower than that of dimer. In our previous experiments, we used the ratio of dimer to monomer around 1:1, which gives us about 42% reaction yield of trimers. Now, we tried excess monomers in the co-evaporation, the yield of triangulene trimers can be further enhanced from 42% to 60%.

We have also tried different coverages and found that coverage will not influence the reaction yield of trimers. Considering all the factors presented above, we used the optimized procedure to maximize the yield of triangulene trimers. The figures below show the typical STM overview images for low and high coverages. We have counted nearly 2000 molecules of different STM overviews for both low and high coverages. The yield of triangulene trimers is calculated by the molecule number of forming triangulene trimers divided by the total number of molecules on the surface, which both give about 60% of reaction yield.

Modification: We have added large STM overviews and more detailed discussion on the experimental conditions and mechanism.

“We have tried different annealing temperatures and found that trimer **2** can be successfully synthesized through surface-assisted dehydration reaction on Au(111) surface with maximized yield upon co-deposition of **3** and **4** then annealing to the temperature of 453 K for 5 min. The other key parameter should be taken into account is the amount ratio of dimer **3** to monomer **4** used in co-deposition. In principle, for ‘2+1’ dehydration reaction the ratio of **3** to **4** is required as 1:1. However, considering the desorption temperature of monomer is lower than that of dimer, the excess amount of monomer **4** in the evaporation will enhance the possibility for ‘2+1’ dehydration reaction. Indeed, we find that upon using excess **4** in the co-evaporation, triangulene trimers can be obtained with high reaction yield of 60% for both low and high coverages, as shown in the overview STM images of Fig.3b,c (nearly 2000 molecules are counted of different overviews at low and high coverages for the statistics).”

2. On the other hand, I found theoretical analysis very weak and careless. The authors rationalise experimental dI/dV maps and spectra using Hubbard mean field method, which depends strongly on employed parameters U/t . It is not clear, why parameter U was chosen 3.5 eV (it seems a small value for carbon/nitrogen) and considered the same for both N and C atoms? One should not rely on a relatively good match with experimental dI/dV maps. Instead, one should try to understand better the magnetic ground and excited states providing a comparative analysis of different U/t parameters. Does it mean that the triazine linker does not have any influence on the magnetic structure and spin excitation spectra? Indeed, what is missing completely is a detailed discussion of possible difference between electronic structure of Tb/Tt products. I suggest to carry out at least some fully relaxed DFT gas-phase calculations with hybrid XC to analysis the atomic and electronic structure of both products including both the ground and first excited states. Such analysis could provide more information about the differences as well as more insight into the effect of triazine linker on the magnetic states and spin excitations. I convinced that these information will be appreciated by colleagues working in the same field and it will extend our understanding of pi-electron magnetism in carbon-based systems.

We thank the reviewer for pointing out the weaknesses of our theoretical analysis. We have addressed Reviewer's comments in below and modified theoretical discussions accordingly to improve our manuscript.

1. We fully agree with the reviewer that the calculation results of mean-field Hubbard model depend strongly on the parameter of U/t . Since in previous calculations, the values of U are usually chosen in the range of 3.0-3.5 eV for polycyclic aromatic hydrocarbons (Phys. Rev. B **31**, 3141 (1985); Phys. Rev. B **35**, 9380(R) (1987)). It has also been shown that mean-field Hubbard calculated results are very close to those obtained by density functional theory for certain values of U/t . In particular, $U/t = 1.3$ can agree best with the results of the generalized-gradient-approximation (Rep. Prog. Phys. **73** 056501 (2010)). Therefore, in our calculation we followed the previous empirical parameter and took $U=3.5\text{eV}$, $U/t=1.3$. We also agree with the reviewer that taking the same U values for atoms C and N in our previous calculations could not be precise enough. Therefore, we have performed spin-polarized DFT calculations for precise calculation of electronic structures of Tb and Tt.

2. We thank the reviewer to pointing out that the discussion on the possible electronic difference between Tb and Tt is missing, which will certainly benefit our manuscript. We have performed spin-polarized DFT calculations on the electronic structures of Tb

and Tt. The results show that $S=3$ is the ground state for both Tb and Tt. The energies of $S=3$ are lower by 13.9 meV and 8.9 meV than that of the first excited state $S=1$ for Tb and Tt, respectively. The calculated frontier orbitals of Tt and Tb both contain six singly occupied and unoccupied molecular orbitals for $S=3$. Their spatial distributions of representative molecular orbitals are displayed in the following figure. For comparison, the energy levels and electronic wave functions of the first excited spin state $S=1$ for Tt and Tb are also illustrated in the following figure. From these results, we can find that the benzene or triazine linker makes no significant difference to the electronic structures of Tb and Tt.

As the figure shown below, we have also compared the calculated results of Tb using mean-field Hubbard model (MFHM) and spin-polarized DFT. Both methods show that $S=3$ is the ground state for Tb and the spin density distributions are also similar, but with slightly excitation energy difference. The $S=3$ ground state is 8.75 meV (13.9 meV) lower than that of excited spin state $S=1$ for MFHM (DFT) calculations, respectively.

Modification: We have revised Fig.4 to compare the electronic difference between Tt and Tb in the manuscript and added the above figures in the supplementary information.

“Mean-field Hubbard model (MFHM) and spin-polarized density functional theory (SP-DFT) have been used to calculate the electronic properties of **Tb** and **Tt**. Both methods indicate that $S=3$ is the ground state for both **Tb** and **Tt**, predicting a ferromagnetic ordering among three triangulene units, which is agree with Lieb’s theory. The MFHM results can be found in the supplementary information (Fig.S14). The DFT calculations reveal that the ground state $S=3$ is with a lower energy of 13.9 meV and 8.9 meV than that of the first excited $S=1$ state for **Tb** and **Tt**, respectively. Fig.4a and b depict the calculated energy levels of **Tb** and **Tt** in the ground state $S=3$. Their frontier orbitals both contain six singly occupied and singly unoccupied molecular orbitals (SOMOs and SUMOs). The quasi-degenerated SOMOs are populated by six ferromagnetically coupled electrons, giving the ground state of $S=3$. The orbital shapes of SOMOs are shown in Fig.4c and d. To experimentally compare the electronic structure of **Tb** and **Tt**, we measured differential conductance (dI/dV) spectra at different positions over **Tb/Tt** using a CO functionalized tungsten tip. As Fig.4e and f shown, the dI/dV spectra measured at the edge corner of **Tb/Tt** (measured position marked by red dot in the inset image) show two pronounced peaks at -0.58 V/-0.5 V and 0.93 V/1.35 V for **Tb** and **Tt**, respectively. These two peaks could be attributed to the presence of the SOMOs and SUMOs. In addition, two ‘shoulder’ features around -1.4 V/-1.8 V and 1.8

V/ 2 V observed in dI/dV spectra measured at the connection corner on **Tt/Tb** could be attributed to several occupied molecular states with close energy (H1) and several unoccupied molecular states (L1), while the substantial rise above -2.0 V may originate from other higher occupied molecular states H2 (The assigned states are marked by dashed rectangle in Fig. 4a and b).

To further resolve the spatial distribution of these molecular orbitals, we have performed dI/dV mappings at the observed resonant energies. As the dI/dV maps measured at -0.5 V and 1.35 V on **Tt** shown (Fig.5a), they are similar and exhibit characteristic nodal patterns localized at the zigzag edges of triangulenes, which agree well with the simulated local density of states (LDOS) maps obtained by superimposing the six orbitals of SOMOs and SUMOs. The dI/dV map obtained at 2 V shows three bright spots in the center, which fit with the simulated LDOS map obtained by superimposing the molecular states of L1 as labeled in Fig.4a. The dI/dV maps acquired at -1.8 V and -2.3 V show faint nodal features and triangle frame along the zigzag edges, which can be reproduced in the simulated LDOS maps using occupied molecular states H1 and H2 (H1 and H2 are marked in Fig.4a). Similarly, the dI/dV maps measured at different resonance energies of **Tb** are also compared with theoretical simulations, which show similar wave patterns to that of **Tt** (Fig.S15). Therefore, we can conclude that both benzene and triazine linkers give the ground state of $S=3$, that is, three triangulene units are ferromagnetic coupled. The electronic structures of the excited spin state $S=1$ for **Tb** and **Tt** can be found in the supplementary information (Fig.S16 and 17).

Response to Reviewer 2:

Comments: The manuscript describes the synthesis of high-spin triangulene trimers on Au (111) surface formed by alkyne trimerization and a novel 2+1 surface-assisted dehydration, and characterization of these trimers by AFM and STM techniques. The obtained results are supported by DFT calculations. The novelty of the presented manuscript lies in the connection of triangulene units via benzene/triazine linker, which creates sublattice imbalance resulting in a high-spin ($S = 3$) system. Presented STM and low-temperature dI/dV spectra support conclusions made by authors about the ground state of the triangulene trimer. However, I am not an expert in the field of on-surface spectroscopy and a reviewer from the field should judge whether the presented STM results suffice to confirm the ground state and antiferromagnetic behavior of the presented molecule. The discussion of the results obtained for triangulene trimer linked via benzene is omitted from the discussion. In my opinion, it would be helpful for the readers of this paper to add a paragraph describing these results, including a direct

comparison with the results for trimer with a triazine linker and making a conclusion about the effect of heteroatoms. Overall, I think that this work possesses novelty and quality of research required for Nature Communications. However, this manuscript contains significant technical issues. After a revision addressing all the points listed below, this manuscript can be re-considered for publication in Nature Communications.

We thank the reviewer for finding our work is novel. We also thank the Reviewer to point out that the discussion on the comparison between Tt and Tb is missing. We have addressed the reviewer's comments point by point in below and the manuscript is modified carefully based on the reviewer's comments.

Comments:

1) Compounds 4, 5, 6 are prepared for the first time (no result in Reaxys&SciFinder). Therefore they must be characterized by another analytical method than just NMR, namely, HR-MS or elemental analysis.

We fully agree with the reviewer. Now besides NMR, we have included HR-MS for all the compounds 4,5,6 in the supplementary information.

Modification: The HR-MS spectra for compounds 4,5,6 are included in Fig.S6, Fig.S9 and Fig.S12.

2) In the synthetic procedure for preparation of compound 4, K₂CO₃ is above the reaction arrow but K₃PO₄ in the text.

We thank reviewer for pointing this error out. It is K₃PO₄ used in the reaction.

Modification: We have corrected this error in the manuscript.

3) NMR is not correctly described several times. For example: 1) For compound 5, the singlet for 2H from phenyl ring is reported as a doublet at 7.41 (based on the amount of H), but this signal cannot be described as a doublet because it is a mixture of 2 signals. Moreover, the coupling J = 12.7 Hz does not fit other coupling constants. 2) For compound 3, methyl groups are described as a doublet with J = 7.5 Hz, but it should be described as two independent singlets. All mistakes of this kind should be corrected.

We are very thankful to the reviewer for pointing these errors out. We have checked all the NMR data, and all the mistakes have been corrected.

Modification: We have corrected all the mistakes of NMR description in the supplementary information.

*4) What are the impurities in the ^{13}C NMR of **5** (Fig. S7)? Authors should assign these impurities.*

Since we purified **5** with petroleum ether, therefore the impurity peaks in the ^{13}C NMR of **5** come from petroleum ether. Now we replaced it with a clean ^{13}C NMR spectrum without such impurity peaks.

Modification: We have exchanged a clean ^{13}C NMR spectrum for Fig.S7.

5) Page 3, Figure 1: Unpaired electrons are part of the π system. I suggest to draw them from the inside of the rings.

We thank the reviewer's suggestion. We have modified Fig.1, in which the unpaired electrons are drawn inside of the rings.

Modification: Fig.1 is modified in the manuscript.

6) Page 4, line 73: Authors use abbreviation Tbs, which does not belong to any compound (probably a typo).

We thank the reviewer to point it out. It should be Tb.

Modification: We have corrected this error in the manuscript.

7) Page 5, Figure 2c: Authors assign bright spot to hydrogen passivation. Why does it happen only with this hydrogen and not with others? A more detailed description in the main text would be helpful.

We thank the reviewer for pointing out our careless omission. Actually, hydrogen passivation could happen at all the reactive carbon sites, not only with this carbon site in Fig. 2c. Since atomic hydrogens can be generated during the dehydrogenation reactions on the surface, it is possible that all the highly reactive carbon sites with unpaired electrons can be passivated by those atomic hydrogens. As the model shown below, the combinations of Clar formulas provide all the possible unpaired electron positions for getting hydrogen atom passivation, which can be randomly passivated during the on-surface reactions. In our experiments, high resolution nc-AFM imaging confirms such random hydrogen passivation process as shown in the following figure.

Modification: To make this point more clearly, we have modified our manuscript in the first paragraph at page 6 as below. We have also included AFM images of triangulene trimers with hydrogen passivation at different carbon sites in the supplementary information.

“The bright spot in the AFM image of Fig.2c is due to an extra hydrogen passivation. Due to the radical character, the carbon sites with unpaired electrons of triangulene show reactivity and can get passivated randomly by hydrogen atoms produced during dehydrogenation reaction on the surface. The observed triangulene trimers with hydrogen passivation at different carbon sites can be found in the supplementary information.”

8) Page 6, line 90: Authors use Fig 3 instead of Fig 3a.

We thank the reviewer to point this error out. We have corrected this error in revised manuscript.

Modification: We corrected it with Fig.3a.

9) Page 8, Figure 4d: How did the authors assign the energy of HOMO? There is no pronounced peak at this energy. It looks like the peak for SOMO is composed of 2 peaks. What is the second peak?

Since the molecule adsorbs on Au(111), the Au surface also has strong density of states at the same energy of HOMO. The superposition of Au states with molecular states leads the resonance at -1.8V to a less pronounced peak. As shown in the following figure, the blue spectrum shows a shoulder at -1.8V, which is absent at bare Au. Additionally, we performed spatial resolved dI/dV maps at this energy, which agree well with the simulated map using HOMO and its nearby states. Combining dI/dV spectra, spatial dI/dV mappings and theoretical calculations, we can attribute the shallow peak at -1.8V to the presence of HOMO and its nearby states as marked in revised fig.4a.

The second peak beside SOMO originates from the electronic state of the tip, which is unrepeatable. We have measured the dI/dV spectrum of triangulene trimers using a different tip, and this feature is missing. As the spectrum shown below (we used larger setpoint by decreasing the tip-sample distance), HOMO and its nearby states can be detected as a 'shoulder' feature and there is no additional peak observed beside SOMO.

Additionally, we found our previous assign of dI/dV peaks to HOMO and LUMO is incorrect. Since the molecule is quite large, there are many states locate at very close energy. In the revised version, we assign the shoulder at 2V to the presence of LUMO and its nearby states; the shoulder at -1.8 to HOMO and its nearby states; and the peak at -2V to other lower occupied states (as marked by dashed rectangle in Fig. 4a and b).

Modification: We have modified Fig.4 accordingly.

10) Figure 2 + Figure 4: It would be clearer to add a size of the scale bar right to the picture instead of having it in the description of the figure.

We thank the reviewer's suggestion.

Modification: We have added the size of scale bars in all the figures.

Response to Reviewer 3:

Comments: Suqin Cheng et al. report on the on-surface synthesis of triangulene trimers via a 2+1 dehydration reaction. They compare reaction schemes yielding triangulene trimers that are connected via a single benzene or triazine ring. These triangulene trimers (Tb and Tt, for benzene and triazine linker, respectively) are exciting as they are supposed to realize nanographenes with high-spin ground states (the connection geometry results in a sublattice imbalance, which implies a certain spin-state via Lieb's theorem), namely six unpaired electrons and $S=3$ in the present case. The synthesis approach is novel, the experiments are carefully carried out and the results are exciting for the broad community working on carbon-based (nanographene) magnetism. I recommend publication in Nature Communications once my comments below have been answered.

We thank the reviewer's positive comments on our work. All the following concerns/comments are fully addressed point by point in our revised version.

1. The authors should compare the experimental dI/dV maps with the theoretically predicted maps also for the other possible spin states (namely $S=1$). This would give more confidence to the assignment of the $S=3$ spin state.

We thank the reviewer's suggestion. We have performed such spin-polarized DFT calculations. The figure below gives the energy levels of the ground spin state $S=3$ and the first excited spin state $S=1$ for both Tb and Tt. We can see that there are only slight energy differences (around few meV) between some energy levels of $S=3$ and $S=1$. Such difference cannot be distinguished in experiments. Therefore, we cannot distinguish the ground state by comparing the experimental dI/dV maps with the theoretical simulated dI/dV maps. As shown below, the observed dI/dV maps agree quite well with the simulated LDOS maps for both $S=3$ and $S=1$ states. Theoretically, both mean-field Hubbard and spin-polarized DFT calculations show that the $S=3$ is the ground state, with an energy around 10 meV lower than that of $S=1$. We assign the ground state to $S=3$ according to such calculations. Experimentally, spin-polarized

measurements may give spin density contrast to distinguish the difference between $S=3$ and $S=1$. Unfortunately, such tedious spin-polarized measurements remain challenging in this field. We tried hard to get such spin-polarized contrast, but failed. In the revised version, we add corresponding discussion to point the experimental determination of the ground state remains to be explored by using spin-polarized measurements.

Modification: We have added discussion in the main manuscript and provide the above figures in the supplementary information.

“Comparing the calculated energy levels of $S=3$ and $S=1$ (Fig.S18), some levels slightly shift with energy difference around few meV. Such difference cannot be distinguished in experiments by comparing the experimental dI/dV maps with the theoretical simulated LDOS maps of $S=3$ and $S=1$ states (Fig.S19). To ambiguously determine the ground state in the experiment, spin-polarized measurements are

required for obtaining the spin density contrast, which is beyond the scope of this work. However, based on the spin-resolved DFT and mean-field Hubbard calculation results, $S=3$ is the ground state with an energy around 10 meV lower than that of $S=1$. Therefore, according to the calculation results, we assign the ground state to $S=3$ and attribute this spin-flip singature³⁷ to the transition from the ground state $S=3$ to the first excited state of $S=1$. The measured magnetic coupling strength is about 10 meV, which also agrees well with theoretical predictions. The results for the spin excitation of **Tb** can be also found in the supplementary information (Fig.S15).”

2. The data in Figure 4: experimental orbital maps were carried out under constant-current conditions, which means that part of the contrast results from the varying tip height. This has minor effect for SOMO and SUMO, but is visible for the higher lying orbitals. Did the authors carry out constant-height experiments and are the theory predictions simulated for constant-current or -height? In addition, how does the experimental dI/dV map of HOMO-1 look like?

We thank the reviewer for this comment. We performed additional dI/dV mapping measurements in the constant-height mode. As the results shown above, most of the key features can be observed in both the constant-height and constant-current measurements, where constant-current maps look clearer since the tip is closer at the molecular boundaries. We simulate the LDOS maps using constant-height mode. The experimental and theoretical simulated maps are shown in the above figure. There is an error in our previous version to assign the dI/dV peaks. Since the molecule is quite large, we found that there are many states locate very closely in energy. The observed peaks cannot be assigned to just one state. As shown in the revised figure 4, we assign the observed dI/dV peaks to the superposition of several states with close energy.

Modification: We have included dI/dV map of higher occupied molecular orbitals in Fig.5. The dI/dV maps measured in the constant-height mode are also included in the supplementary information for comparison.

3. The spin-flip transitions (Fig. 4f): what transition does it correspond to and what is the predicted energy of this? How would this change if the ground state was $S=1$?

We attribute this spin-flip signature to the transition from the ground state $S=3$ to the first excited state $S=1$ according to our calculations. Both Mean-field Hubbard and DFT calculations give that the $S=3$ is the ground state, with an energy around 10 meV lower than that of the first excited $S=1$ state. If we assume the ground state is $S=1$, it would be possible this transition is from the $S=1$ ground state to the first excited $S=3$ state. As discussed in the reply of the reviewer's first question, we need spin-polarized measurements to experimentally determine the ground state.

4. What are the differences in the theory predictions for Tt and Tb molecules? Presently, the manuscript silently implies that all the spin transitions are identical. I understand that the ground state should be $S=3$ in both cases, but what about the predicted spin-flip energies? Experimentally they seem very similar.

We have performed additional spin-polarized DFT calculations for both Tt and Tb triangulene trimers to determine the spin transition energy difference. The excitation energies from $S=3$ to $S=1$ are 8.9 meV and 13.9 meV for Tt and Tb, respectively. However, our dI/dV spectra indicate that the flip energy is almost the same for both Tt and Tb, with a value around 10 meV. Due to the presence of Au, the spin-flip features are very broad, and thus we cannot precisely determine the flip energy to distinguish such tiny energy difference.

Modification: We have added a new paragraph to discuss the electronic state difference for Tt and Tb in the revised version.

5. Do all the SOMO and SUMO orbitals have identical energy or are there some small splittings (according to the calculations)?

According to mean-field Hubbard calculations, all the SOMOs and SUMOs are degenerate without energy splitting. According to spin-polarized DFT calculations, we found there are small energy splittings for both Tb and Tt, which is attributed to structural relaxations. In mean-field Hubbard model, all the carbon-carbon bonds are identical, while the bonds are slightly different in DFT calculations after relaxation.

Modification: We added a figure in the supplementary information to compare the mean-field Hubbard and DFT calculations.

Minor:

- Please add experimental parameters for the AFM experiments, e.g. cantilever oscillation amplitude. Secondly, I wanted to check if the STM experiments are carried out with a CO terminated or a metallic tip?

Modification: All the experimental parameters for AFM measurements are added in the section of experimental method. The STM experiments are also carried out with a CO tip. This information is also added in the method section.

REVIEWER COMMENTS

Reviewer #1 (Remarks to the Author):

I think the authors have responded sufficiently to all the reviewer's comments. The changes made in the manuscript significantly improved its quality of the article. Therefore I can recommend for publication in Nature comm.

Reviewer #2 (Remarks to the Author):

All comments from my side have been addressed satisfactorily. In case the other reviewers are satisfied as well, I recommend publication as is.

Reviewer #3 (Remarks to the Author):

I am satisfied with the authors' reply, they have fully answered my questions. I recommend publication of the manuscript.